# The Artificial Neural Network as a Diagnostic Tool of the Risk of *Clostridioides difficile* Infection among Patients with Chronic Kidney Disease

**DOI:** 10.3390/jcm12144751

**Published:** 2023-07-18

**Authors:** Jakub Stojanowski, Andrzej Konieczny, Łukasz Lis, Weronika Frosztęga, Patrycja Brzozowska, Anna Ciszewska, Klaudia Rydzyńska, Michał Sroka, Kornelia Krakowska, Tomasz Gołębiowski, Zbigniew Hruby, Mariusz Kusztal, Magdalena Krajewska

**Affiliations:** 1Department of Nephrology and Transplantation Medicine, Wroclaw Medical University, Borowska 213, 50-556 Wroclaw, Poland; andrzej.konieczny@umw.edu.pl (A.K.); weronika.frosztega@umw.edu.pl (W.F.); patrycja.brzozowska@student.umw.edu.pl (P.B.); klaudia.rydzynska@student.umw.edu.pl (K.R.); korneliakrakowska006@gmail.com (K.K.); tomasz.golebiowski@umw.edu.pl (T.G.); mariusz.kusztal@umw.edu.pl (M.K.); magdalena.krajewska@umw.edu.pl (M.K.); 2Department of Nephrology with Transplantation and Internal Medicine Subunits, Regional Specialistic Hospital, Kamienskiego 73a, 51-124 Wroclaw, Poland; lislukasz@ymail.com (Ł.L.); aniaa.ciszewska@gmail.com (A.C.); michal.sroka@wssk.wroc.pl (M.S.); zhruby@wssk.wroc.pl (Z.H.)

**Keywords:** artificial intelligence, machine learning, *Clostridioides difficile*, chronic kidney disease

## Abstract

The majority of recently published studies indicate a greater incidence and mortality due to *Clostridioides difficile* infection (CDI) in patients with chronic kidney disease (CKD). Hospitalization, older age, the use of antibiotics, immunosuppression, proton pump inhibitors (PPI), and chronic diseases such as CKD are responsible for the increased prevalence of infections. The aim of the study is to identify clinical indicators allowing, in combination with artificial intelligence (AI) techniques, the most accurate assessment of the patients being at elevated risk of CDI.

## 1. Introduction

Numerous publications indicate an increased number of complications, prolonged hospitalization time, mortality, and treatment costs due to *Clostridioides difficile* infection (CDI) in patients with chronic kidney disease (CKD).

In CDI the etiological factors are toxins A and/or toxin B and binary toxin produced by toxicogenic strains of *Clostridioides difficile* (CD), developing in the intestine mainly as a result of intestinal microbiota disorders, which accompanies the use of antibiotics during long hospitalization. The main symptoms of CDI are fever, abdominal pain and watery diarrhea of varying severity, which in itself exposes the patient to water and electrolyte disturbances [1], but the spectrum also includes pseudomembranous colitis, which is associated with the most severe form of diarrhea associated with antibiotic therapy.

Machine learning allows one to imitate a simple decision-making process and performs data classification based on previously calculated dependencies. These dependencies come from supervised learning (i.e., a person gives the correct labels for existing and available data, and the role of the program is to match the so-called hyperparameters to show the greatest possible discriminatory ability on the various input data that will be used to evaluate the models). Overcomplicating the model risks overfitting the model to the data and completely ineffective prediction on new data. Too little fit or too little complexity makes the model comparable to, or even worse than, the random classifier.

The aim of our work was to construct an effective predictive model with machine learning in order to determine whether a patient is at risk of developing *Clostridioides difficile* infection during hospitalization lasting more than 72 h. The main objective was to classify the patient, at some point, into a risk group and give him special care and a special examination to investigate complications.

The technique used allowed us to build predictive models that may be used in risk stratification by classification, screening testing, or predicting the course of the disease in patients with various diseases. It is crucial to provide an appropriate database, containing the parameters measured for all patients. It is also necessary to supervise the model learning process by properly labeling records (i.e., assigning a class, for example “no infection” and “infection”). This is how we implemented supervised machine learning methods.

Artificial intelligence is a flexible tool that may be implemented in various clinical applications. One is to assess the risk of *Clostridioides difficile* infection at the time of admission. This has therapeutic and economic implications. Similar models can be widely used in various issues in the field of medicine.

## 2. Materials and Methods

### 2.1. Data Collection

The study was focused on the CDI risk prediction in 252 patients with CKD. admitted to two nephrological departments between 1 January 2016 and 31 December 2020. Technically, the classifier was based on artificial intelligence techniques, in particular the random forest classifier. Initially, a set of parameters was selected to evaluate different predictive models and choose the one with the higher predictive value, considering the different input fields. The database of 252 patients was randomly divided into two subsets: training and testing in a ratio of 80:20, with 201 and 51 patients, respectively (Table 1).

### 2.2. Patient Qualification Criteria

The inclusion criteria were the presence of diarrhea, with more than three stools per day, and abdominal pain or fever, which developed more than 72 h after admission, among adult patients [1]. In all patients, a rapid enzyme cassette immunoassay was performed, detecting the antigens of toxins A and B of CD in stool (TOX A/B QUIK CHEK^®^; Techlab, Blackburg, VA, USA).

The patients were assessed in Norton Scale (ANSS) and classified as care class on a scale of 1 to 4, where 1 indicates a self-care patient, 2 a partial care patient, 3 a complete care patient, and 4 a critical care patient.

The exclusion criteria were missing medical history, length of stay less than three days and transfer from another hospital. None of the patients included in the study used laxatives or tube feeding.

### 2.3. Statistical Scoring

Several descriptive statistics are used to evaluate the binary classifier, which is the random forest classifier. Using too few statistics runs the risk of overestimating the classifier and being overly optimistic. To demonstrate the effectiveness of our model and to guarantee genuine practical application, we used several statistics to describe the performance. The model with the best statistics was finally saved and evaluated in detail. For the statistical description of the model, the following parameters were used:

Accuracy is the ratio of correct matches or classifications to all predictions made, which is the degree to which the model’s predictions are close to the true value. Accuracy is a value that is sensitive to the balance of the set. If there is a predominance of one type in a dataset, there is a risk that accuracy will not be a reliable parameter. It is then required to use additional parameters that will guarantee reliable performance of the predictive model. Accuracy is a parameter that is very intuitive and easy to interpret and compare classifiers. If the set contains more elements of one class, the classifier can achieve high accuracy even though it will randomly classify the elements. This case is prevented, apart from other measures, by the use of the area under the receiver-operator curve, which is 0.5 for the random classifier. Our dataset is definitely unbalanced, so we expected to use several statistics. However, the dataset contains enough minority class elements to ensure diversity and efficiency as measured by the other statistics described below.

The area under the receiver-operator curve (AUROC) is a basic parameter that determines the discriminating ability of a binary classification model. The area under the receiver-operator curve takes values from 0.0 to 1.0, where the value 0.5 applies to the random classifier. In other words, a model with an area under the ROC of less than 0.5 is a worse predictor than a coin toss.

Precision is the computer science equivalent of a positive predictive value and is defined as the ratio of true hits to all positive calls.
$$\mathrm{Precision}=\frac{\mathrm{TP}}{\mathrm{TN}+\mathrm{FP}}$$

Recall is equivalent to sensitivity and is the ratio of true positives to the sum of true positives and false negatives.
$$\mathrm{Recall}=\frac{\mathrm{TP}}{\mathrm{TP}+\mathrm{FN}}$$

The F1 result is the harmonic mean of precision and memory. F1-score is a stat especially useful in unbalanced sets due to classes. However, it has a limitation that can be circumvented by using a broader and more complex statistic, such as the Matthews correlation coefficient. F1 score takes values from 0.0 to 1.0. If the model misclassifies all positive samples, the F1 score is 0.0. In contrast, 1.0 is achieved for a perfect classification with no false positives or false negatives.
$$F1\;\mathrm{Score}=\frac{2}{{\mathrm{Precision}}^{-1}+{\mathrm{Recall}}^{-1}}=\frac{2\mathrm{TP}}{2\mathrm{TP}+\mathrm{FP}+\mathrm{FN}}$$

Abbreviations: TP—true positives, FP—false positives, TN—true negatives, and FN—false negatives.

The Matthews correlation coefficient (MCC) is a parameter that reaches a high value if the model performs well enough in all four fields of the confusion matrix [2]. The use of all statistics increases the evaluation time of many models but strengthens the proof of the effectiveness of the predictive model. MCC is an indicator whose value is high with appropriately correct classification, both positive and negative, in contrast to the previously quoted statistical measures. The MCC value ranges from −1 to +1, with extreme values being reached for perfect misclassification and perfect classification, respectively. MCC = 0 corresponds to a random classifier.

In the case of our database, three statistical measures are sufficient: accuracy, F1-score, and MCC [2].

### 2.4. Random Forest Classifier

A single decision tree classifies a patient data record on the basis of multiple divisions and selecting the lower node due to the fulfillment of the condition contained in the higher node. A random forest classifier is a classifier composed of multiple decision trees trained on randomly different subsets of the training set with an aim to reduce variance. A random forest classifier divides the training set into bootstraps, which are used to build decision trees. When we query such a classifier, the individual decision trees return their results, which are ultimately averaged based on the majority of occurrences. 

A single decision tree node splits a set, based on a condition contained in that node. In our case, it is a minority relation. If any of the splits leads to a set of elements with one type of label, we say that there is zero impurity and no more splits are made at that point. We expect to achieve less class impurity with each node. The randomness of a decision tree means that a random set of variables is selected from the input variables to generate a single tree. In this way, the amount of data encoded in the tree is reduced and overfitting to the training data and insensitivity to the testing data are avoided. The use of a set of decision trees allows for covering the training set and better performance. RFC is a classifier insensitive to the scaling of input variables. In addition, RFC is characterized by greater stability in relation to the number of input variables, which means that, for example, a redundant input variable does not have to interfere with the generation of a forest with good performance. It can then be easily eliminated, and the entire model optimized for clinical use. Gini feature importance is the sum of average impurity declines in the whole tree. The greater the Gini feature importance, the greater the contribution of a given variable to the clarification of sets and the greater the number of significant divisions.

### 2.5. Algorithm

From the original database, containing 23 parameters, tables containing selected columns were selected recursively. In this way, subsets of input variables were obtained, on which the developed program built a random forest classifier model, then optimized it and saved statistical scoring (5-cross validation by MCC) in logfile. Eventually, we selected the best set of input variables and the model that guaranteed the best performance in predicting the occurrence of CDI. The minimum size of the input data was 3 parameters, at which the program recursively goes to a new subset of the input data. We obtained the optimal execution of the program by parallelizing the calculations.

### 2.6. Feature Importances

An additional statistic describing the contribution of individual input variables to the final predictive ability is typical for random forest and is called Gini feature importance which value directly describes feature relevance. Gini feature importance is the higher for a given parameter, the greater the share of the given parameter in the prediction. In other words, if in a given model feature has high Gini importance, then the model without this variable will have significantly worse performance. In case of random forest, Gini feature importance also means that the given parameter is responsible for quite a lot of important splits of the input data into the final classes.

## 3. Results

This was a retrospective study based on data of 252 patients, with both clinical signs and proved CDI, randomly divided into two sets (i.e., training and testing) (Table 2). Based on training data, we built models that we evaluated in five-cross validation for MCC.

The top random forest model made it possible to assess the risk of CDI in patients from testing set hospitalized for more than three days with very high accuracy (98.04%), precision (0.9809) and recall (0.9804). The area under the receiver-operator curve (AUROC) achieved 0.9545 (95% CI 0.8665-1.0) and MCC 0.94 (Figure 1). These statistics refer to the testing set. The AUROC value of 0.9545 suggests that the model is close to the ideal classifier, thus indicating a very clear discrimination power. Additionally, the MCC at the level of 0.94 should be interpreted as a very good measure of the classifier, which turned out to perform very well in each of the four fields of the confusion matrix. The variables were selected on the basis of the training set. Parameters considered, at the time of hospital admission, were the length of antibiotics use before CDI in days, status of ER stay before admission, Norton scale, care class, BMI with importance of 46.99%, 8.55%, 23.28%, 8.13%, and 13.05%, respectively. This means that the length of antibiotics use as a determinant variable occurred in almost half of the conditions in the nodes and thus accounts for almost half of all divisions of the input set in relation to the final class “no infection” or “infection”. Secondly, in the prediction and separation of the input set, the Norton scale and then the body mass index (BMI) are important.

In each class, the model achieved a precision of 0.98 and 1.00 for non-CDI and CDI class, respectively, recall of 1.00 and 0.91 and F1-score of 0.99 for no-CDI and 0.95 for CDI class (Table 3). In the table we show the results of the precision, recall and F1 score statistics for the individual classes “no-CDI” and “CDI” regarding the absence of infection with *Clostridioides difficile* or infection, respectively.

The macro average is the arithmetic average of the respective precision, recall, F1-score values for both “no-CDI” and “CDI” classes. The weighted average is the weighted average of the respective precision, recall, and F1-score values, considering the number in both classes. The model has excellent sensitivity in rejecting CDI infection during hospitalization. It also has perfect positive predictive power in predicting CDI infection. Our model detects true positives slightly more often for the no-CDI class, which results from the higher F1-score for this class. Nevertheless, the remaining statistical measures remain at a high level, indicating the effectiveness of the model developed by us. Detailed leave-one-out cross validation was performed on the test set and showed an average AUC of 0.94 (Figure 2).

## 4. Discussion

To the best of ourknowledge, this is the first study using artificial neural network as a diagnostic tool to establish risk factors of CDI, among patients with CKD.

In the study by Lis et al. the length of antibiotic therapy and the Norton class were among the key parameters associated with the risk of CDI [3]. In our experience, these factors also turned out to be the most important, which in a way indicates the convergence of traditional statistical analysis with modeling based on machine learning.

Normal intestinal microbiota practically protects against CDI, while disorders resulting from the extensive use of antibiotics may lead to overgrowth of CD strains, producing toxins responsible for diarrhea, including pseudo-bloc enteritis. Different groups of antibiotics are significantly associated with the development of CDI and the risk varies depending on the antibiotic or chemotherapeutic agent and the duration of its use, which in our model is the most important input parameter [4].

Also, ER stay before admission as risk factor has been considered by other authors as an independent factor, especially regarding infections among patients who did not used antibiotics or medical care, apart from their stay at the ER [5,6,7,8]. Those data correlate with our present results.

The Norton scale combines five parameters: general physical condition, mental state, physical activity, mobility and abstinence. A lower score is associated with less activity and greater dependence on the environment and greater exposure to pathogenic factors [9]. It is considered as a predictor for in-hospital mortality on internal medicine departments [10]. The present study revealed increased CDI among patients with lower Norton scale score.

Greater dependence of patients on outside help applies to patients particularly exposed to broad antibiotic therapy, elderly patients, and the chronically or seriously ill. Care class is a fairly widely interpreted and used indicator that can, however, determine whether a patient is at risk of CDI [11].

In some studies, weight disorders in the form of being underweight or having extreme obesity, with a BMI over 35, are associated with an increased risk of CDI [12,13]. However, according to recent metanalysis BMI was found to have negative correlation with CDI [14]. Our study results indicate BMI as a relevant risk factor for CDI development. This topic needs the further investigation in order to establish possible relation.

With the help of machine learning, we can effectively predict whether a patient hospitalized in the ward is likely to develop CDI. The assessment is possible on the basis of simple indicators, assessed at the time of admission, and they do not require costly tests or analyses.

Ranking based on the MCC is conducive to finding a model that is reliable in all four fields of the confusion matrix. However, if we were looking for a model whose task would only be to accurately exclude or confirm a disease, it is possible that a model with a worse MCC would be more advantageous, especially when it depends more on the sensitivity or specificity of the classifier.

Our study had several strengths. It is remarkable that no study before has investigated role of artificial neural network in CD development among patients with CKD. In addition, relatively large study group (*N* = 252) underwent artificial intelligence-based neural network investigation.

A limitation of our study was the two-center study population, which could be expanded in the future and the introduction of a CDI risk assessment upon adoption as common practice could be considered. The currently used scales allow to identify risk groups, while the model compiled from various relationships between variables allows for precise indication to which group the patient belongs. The larger the group of patients, the more reliable the model will be.

## 5. Conclusions

The study supports the usefulness of AI in a reliable prediction of CDI risk. In our study, length of antibiotic use before CDI, ER stay before admission, Norton scale score, care class, and BMI were considered as risk factors for CDI development. AI allows for the identification of patients at risk, who should be carefully monitored for possible complications and implementation of treatment preventing the occurrence of CDI. The two-center machine learning study to assess the risk of *Clostridioides difficile* infection may be extended in the future to include new patient data from other centers as well. Artificial intelligence techniques are scalable solutions that can be used in various fields of medicine which we have shown in our previous publications on this subject [15,16,17].

Similar machine learning models used to predict the risk of *Clostridioides difficile* infection can be used in conjunction with other models based on the same technique to comprehensively manage patient therapy and optimize hospitalization both in terms of treatment effectiveness and minimizing generated costs.

## Figures and Tables

**Figure 1 jcm-12-04751-f001:**
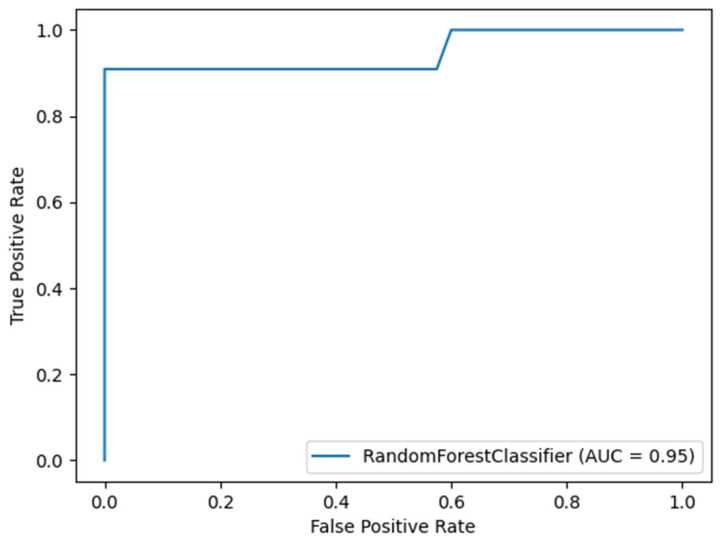
Random forest classifier with given input variables: length of antibiotics used before CDI, in days, status of ER stay before admission, Norton scale, care class, BMI showed very good ability to discriminate patients who developed CDI.

**Figure 2 jcm-12-04751-f002:**
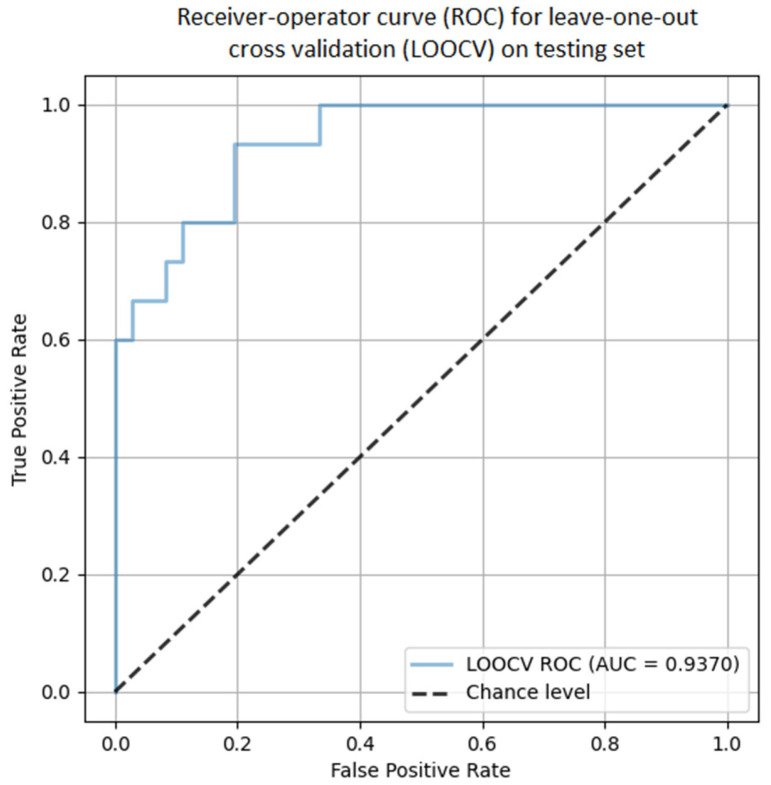
Due to the risk of bias due to the small number of samples, a leave-one-out cross validation analysis was performed which resulted in an averaged AUC of 0.9370.

**Table 1 jcm-12-04751-t001:** Patients’ baseline characteristic.

Variable	Mean ± SD (Min–Max)
Age	65.57 ± 16.47 (22–96)
LOS [days]	14.93 ± 14.66 (3–105)
HD treatment	20 (7.9%)
CDK Stage 1	26 (10.3%)
CDK Stage 2	35 (13.9%)
CDK Stage 3	64 (25.4%)
CDK Stage 4	60 (23.8%)
CDK Stage 5	67 (26.6%)
Serum creatinine concentration at admission [mg/dL]	2.88 ± 2.70 (0.46–23.28)
Serum urea concentration at admission [mg/dL]	100.02 ± 73.34 (9–450)
Serum Albumin concentration at admission [g/dL]	3.42 ± 0.69 (1.5–5.72)
Tumor presence	24 (9.5%)
Number of antibiotics used before CDI-0	143 (56.7%)
Number of antibiotics used before CDI-1	49 (19.4%)
Number of antibiotics used before CDI-2	43 (17.1%)
Number of antibiotics used before CDI-3	11 (4.4%)
Number of antibiotics used before CDI-4	4 (1.6%)
Number of antibiotics used before CDI-5	2 (0.8%)
Length of antibiotics use before CD [days]	5.60 ± 8.42 (0–60)
PPI use	121 (48.0%)
Probiotics use	47 (18.7%)
Statins use	75 (29.8%)
Immunosuppression	86 (34.1%)
Diabetes	72 (28.6%)
The Padua prediction score-0	51 (20.2%)
The Padua prediction score-1	81 (32.1%)
The Padua prediction score-2	29 (11.5%)
The Padua prediction score-3	14 (5.6%)
The Padua prediction score-4	17 (6.7%)
The Padua prediction score-5	31 (12.3%)
The Padua prediction score-6	15 (6.0%)
The Padua prediction score-7	9 (3.6%)
The Padua prediction score-8	3 (1.2%)
The Padua prediction score-9	2 (0.8%)
Gender 0	122 (48.4%)
ER stay before admission	128 (50.8%)
Norton scale	16.54 ± 3.49 (5–20)
Care class-1	99 (39.3%)
Care class-2	95 (37.7%)
Care class-3	58 (23.0%)
BMI	25.98 ± 5.48 (17–47)
Presence of AKI at admission 1	56 (22.2%)
CDI	72 (28.6%)

Abbreviations: LOS—length of stay; HD—hemodialysis; CKD—chronic kidney disease; CDI—*Clostridioides difficile* infection; ER—emergency department; AKI—acute kidney injury; PPI—proton pump inhibitor; BMI—body mass index.

**Table 2 jcm-12-04751-t002:** Patients’ baseline characteristic after dividing into two sets.

Variable	Training Set *N* = 201	Testing Set *N* = 51
	Mean ± SD (Min–Max)	Mean ± SD (Min–Max)
Age	65.12 ± 16.87 (22–96)	67.35 ± 14.84 (36–94)
LOS [days]	14.82 ± 14.95 (3–105)	15.35 ± 13.59 (3–59)
Hemodialysis treatment	15 (7.5%)	5 (9.8%)
CDK Stage 1	23 (11.4%)	3 (5.9%)
CDK Stage 2	26 (12.9%)	9 (17.6%)
CDK Stage 3	53 (26.4%)	11 (21.6%)
CDK Stage 4	43 (21.4%)	17 (33.3%)
CDK Stage 5	56 (27.9%)	11 (21.6%)
Serum creatinine concentration at admission [mg/dL]	2.93 ± 2.86 (0.46–23.28)	2.68 ± 1.94 (0.5–9.63)
Serum urea concentration at admission [mg/dL]	97.78 ± 72.14 (9–450)	108.86 ± 78.02 (22–429)
Serum Albumin concentration at admission [g/dL]	3.4 ± 0.73 (1.5–5.72)	3.49 ± 0.52 (2.2–4.5)
Tumor presence	21 (10.4%)	3 (5.9%)
Number of antibiotics used before CDI-0	114 (56.7%)	29 (56.9%)
Number of antibiotics used before CDI-1	41 (20.4%)	8 (15.7%)
Number of antibiotics used before CD-2	33 (16.4%)	10 (19.6%)
Number of antibiotics used before CD-3	8 (4.0%)	3 (5.9%)
Number of antibiotics used before CD-4	3 (1.5%)	1 (2.0%)
Number of antibiotics used before CD-5	2 (1.0%)	0 (0.0%)
Length of antibiotics use before CDI [days]	5.39 ± 8.37 (0–60)	6.43 ± 8.65 (0–30)
PPI use	99 (49.3%)	22 (43.1%)
Probiotics use	39 (19.4%)	8 (15.7%)
Statins use	62 (30.8%)	13 (25.5%)
Immunosuppression	68 (33.8%)	18 (35.3%)
Diabetes	58 (28.9%)	14 (27.5%)
The Padua prediction score-0	39 (19.4%)	12 (23.5%)
The Padua prediction score-1	69 (34.3%)	12 (23.5%)
The Padua prediction score-2	24 (11.9%)	5 (9.8%)
The Padua prediction score-3	12 (6.0%)	2 (3.9%)
The Padua prediction score-4	10 (5.0%)	7 (13.7%)
The Padua prediction score-5	24 (11.9%)	7 (13.7%)
The Padua prediction score-6	13 (6.5%)	2 (3.9%)
The Padua prediction score-7	6 (3.0%)	3 (5.9%)
The Padua prediction score-8	2 (1.0%)	1 (2.0%)
The Padua prediction score-9	2 (1.0%)	0 (0.0%)
Gender 0	101 (50.2%)	21 (41.2%)
ER stay before admission	97 (48.3%)	31 (60.8%)
Norton scale	16.54 ± 3.56 (5–20)	16.55 ± 3.25 (7–20)
Care class-1	79 (39.3%)	20 (39.2%)
Care class-2	78 (38.8%)	17 (33.3%)
Care class-3	44 (21.9%)	14 (27.5%)
BMI	25.88 ± 5.69 (17–47)	26.33 ± 4.62 (17–35)
Presence of AKI at admission	43 (21.4%)	13 (25.5%)
CDI	58 (28.9%)	14 (27.5%)

**Table 3 jcm-12-04751-t003:** Performance statistics for individual classes: non-CDI and CDI.

	Precision	Recall	F1-Score
Non-CDI	0.98	1.00	0.99
CDI	1.00	0.91	0.95
Accuracy			0.98
Macro average	0.99	0.95	0.97
Weighted average	0.98	0.98	0.98

## Data Availability

Data are contained within the article.

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
