# Peer review of "The Artificial Neural Network as a Diagnostic Tool of the Risk of Clostridioides difficile Infection among Patients with Chronic Kidney Disease"

_jcm, 2023, doi:10.3390/jcm12144751_

Round 1

Reviewer 1 Report

The method of artificial intelligence is an emerging technology that is not proven. For this reason this is an important article.

The introduction should discuss if this model has been employed in other disease states. If it has not been employed before there should be a comment that this is an unproved approach.

There was no exclusion criteria. There always needs to be exclusion criteria to  define the population such as laxative usage and use of tube feeding.

I am not qualified to comment on the random forrest classifier used to qualify the results. Please ask a statistician to review this methods section.

The results are confusing because these statistical methods are not routinely utilized. More discussion is needed to exclaim the significant results. 

The results are not informative but the method is key. Please have the methods reviewed by a statistician and have the authors describe the statistical results in greater extent.  

Author Response

Thank you for your valuable comments.

Similar models have been used in other issues, as described in already published articles. Previously we published papers about implementation of artificial intelligence in predicting renal remission in lupus nephritis, kidney transplant outcome and IgA nephropathy.

We supplemented the information about the exclusion criteria.

We have supplemented the description of the methods and results in order to better understand the statistical measures quoted. The described parameters as Accuracy, Recall, precision, F1-score and MCC are based on the classic confusion matrix. We have included an appropriate explanation of these parameters in a general sense and in relation to our results.

Reviewer 2 Report

The Artificial neural network as a diagnostic tool of the risk of 2 Clostridioides difficile infection among patients with chronic 3 kidney disease. 

The manuscript presents a retrospective study aimed at constructing a predictive model using machine learning techniques to determine the risk of developing Clostridioides difficile infection (CDI) in hospitalized patients. The authors focused on 252 patients with CKD admitted to two nephrological departments between January 2016 and December 2020. The study employed a random forest classifier and various statistical measures to evaluate the model's performance. The results showed high accuracy, precision, recall, and an area under the receiver-operator curve (AUROC) of 0.9545. The manuscript also discussed the importance of selected input variables and their relevance to CDI risk. While the study has strengths, such as being the first to use artificial neural networks for CDI risk assessment in CKD patients, it also has limitations. The two-center study population and the need for further expansion and validation are important considerations. Overall, the manuscript provides valuable insights into CDI risk prediction, but there are areas that require improvement and clarification.

The manuscript mentions the inclusion criteria based on the presence of diarrhea, but it lacks clarity on the diagnostic methods used to confirm CDI. Providing more information on the diagnostic process would improve the understanding of the study.

The manuscript introduces several statistical measures, such as accuracy, AUROC, precision, recall, F1-score, and the Matthews correlation coefficient (MCC). However, the authors do not thoroughly explain the interpretation and implications of these measures. Providing a clear explanation of each measure and their significance would enhance the readers' understanding of the model's performance.

While the authors briefly mention the limitations of the study, they should elaborate on the impact of the two-center study population and the potential for future expansion. Additionally, the authors should discuss the generalizability of the findings and address the need for external validation.

It appears that the manuscript could benefit from language editing.

It appears that the manuscript could benefit from language editing.

Author Response

Thank you for recognizing our contribution and work.

The diagnostic method used to confirm Clostridioides difficile infection was added.

We have introduced an appropriate explanation and formulas with interpretation. We've linked them to our study in the discussion section. Thank you for highlighting the potential of our work. We raised the topic of further development with reference to our previous achievements in the field of using machine learning in medicine.